# Anterior, Posterior, and Thickness Cornea Differences after Scleral Lens Wear in Post-LASIK Subjects for One Year

**DOI:** 10.3390/healthcare11222922

**Published:** 2023-11-08

**Authors:** Maria Serramito, Ana Privado-Aroco, Gonzalo Carracedo

**Affiliations:** 1Department of Optometry and Vision, Faculty of Optics and Optometry, Complutense University of Madrid, 28037 Madrid, Spain; aprivado@ucm.es (A.P.-A.); jgcarrac@ucm.es (G.C.); 2Ocupharm Research Group, Faculty of Optics and Optometry, Complutense University of Madrid, 28037 Madrid, Spain

**Keywords:** scleral lens, post-LASIK, corneal curvature, corneal thickness

## Abstract

The aim of this study is to analyze the anterior and posterior corneal surface shape and the corneal thickness difference outcomes between before and after scleral lens (ScCL) wear in post-LASIK ectasia subjects for one year. Twenty eyes with post-LASIK ectasia wearing scleral lenses were evaluated in a visit before contact lens and after 1, 6, and 12 months. The study variables analyzed included the apex, nasal, temporal, inferior, and superior corneal thickness; the anterior and posterior surface corneal at corneal diameters of 8, 6, 4, and 2 mm, and high-contrast visual acuity. A statistically significant increment of corneal thickness (*p* < 0.05) was observed in the inferior area after 6 months and in the superior area in the 12-month follow-up after wearing ScCLs. The anterior corneal curvature presented a flattening and a statistically significant steepening (*p* < 0.05) in the central and peripheral radii, respectively, after one year. The posterior corneal curvature showed a significant (*p* < 0.05) steepening, which mainly affected the central region after one year. Despite these changes, high-contrast visual acuity with ScCL correction remained at the same values. The prolonged use of scleral lenses in post-LASIK subjects showed significant changes in the corneal curvature and thickness. These outcomes recommend more detailed and periodic topographic and vision quality checks to monitor the wear in ScCL patients.

## 1. Introduction

Post-LASIK ectasia is defined as a progressive structural deformation of the cornea after having undergone laser in situ keratomileusis surgery (LASIK) without complications. This ectasia arises from a deformation of the ablated cornea, affecting both the anterior and posterior curvatures and causing optical refractive instability [1,2]. It is related to defective topography corneal preoperative, altered biomechanical properties, and excessive laser ablation [3].

This ectasia may be the most characteristic biomechanical disease after ophthalmologic surgery [2], with an incidence of 0.033% over eight years, as reported by Bohac and associates in 2018 in a retrospective case series [4]. Currently, the treatment for this condition continues to be a therapeutic refractive surgeon’s challenge. Some of the treatments used are: rigid gas-permeable contact lenses [1,5], corneal crosslinking (CXL) [2,6], crosslinking with simultaneous topography-guided photorefractive keratectomy [7], implantations of an intrastromal corneal ring segment (ICRS) [8,9], and corneal transplantation [10]. Nevertheless, more studies are needed to ensure the best treatment in each case [9].

The post-LASIK ectasia and keratoconus ectasia pathogenesis have histopathological, ultrastructural, and biochemical differences [11,12]. In fact, this corneal refractive surgery complication could be considered equivalent to a biological alteration of the composition of the interfiber material, which occurs in biomechanically weakened corneas at times when the structural stability is altered by excimer laser refractive surgery [11,13].

Clinical signs of this ectasia include a mixture of different signs, such as anterior and posterior corneal steepening, stromal thinning, corneal aberrations, a progressive increase in irregular astigmatism, myopia, and loss of corrected distance visual acuity (CDVA) [11,14,15]. All this results in a variety of symptoms ranging from mild to severe vision deficiency, glare, and ghosting, among other visual manifestations [16]. The main concern for the practitioner and patient with post-LASIK corneal ectasia is irreversible vision loss in a previously normal eye with adequate corrected vision prior to surgery [11].

Scleral lenses (ScCLs) have experienced increased usage in recent years because of their improved vision correction results and also due to the improved comfort level they provide [17]. These lenses continue to be a fundamental solution for the visual compensation of patients with irregular astigmatism [18], such as corneal ectasia. In addition, they provide successful visual and therapeutic solutions when all other treatment modalities fail [19]. ScCLs have a large diameter and are contact lenses made of a rigid gas-permeable material. The lens support area is intended to rest on the sclera and form a dome on the corneal curvature, known as a post-lens tear film. The post-lens tear film is responsible for neutralizing corneal astigmatism and the majority of higher-order aberrations [20].

At present, ScCLs may be used to improve and minimize the symptoms of dryness, acting as a therapeutic treatment for patients with dry eyes [21]. The development of dry eye after LASIK is one of the most common postoperative conditions following ophthalmic surgery [22]. However, ScCLs may produce adverse effects after long periods of time without breaks and correct guidelines [18], such as an increase in the corneal thickness, indicative of hypoxic stress-induced edema [23]. It is, therefore, essential to quantify variations in the corneal thickness and curvature to know their effects on corneal health, acuity, and contrast sensitivity [24].

Some short-term studies have analyzed changes in the corneal thickness [25], anterior curvature [26], and posterior curvature [27] in ScCL wearers with keratoconus. However, in the scientific literature, the possible corneal changes induced by the use of this type of rigid contact lenses with post-LASIK ectasia after one year of lens wear has not previously been reported. Therefore, the goal of this work was to assess the anterior and posterior surface curvature and corneal thickness changes after ScCL wear for one year in post-LASIK ectasia subjects.

## 2. Materials and Methods

This prospective and long-term study included twenty subjects with irregular cornea after refractive surgery by laser in situ keratomileusis (LASIK). The participants recruited were patients of the clinic Optometry Clinic of the Complutense University of Madrid.

The patients presented corneal surface irregularities, specifically cornea ectasia, after they had undergone LASIK surgery for the correction of refraction. Furthermore, they presented refractive problems that could not be solved by ophthalmic lenses or soft contact lenses because of unsatisfactory visual quality. The exclusion criteria used were atopy, allergies, and ocular diseases, since they could cause changes in the ocular parameters analyzed. By avoiding these exclusion criteria, changes in the corneal curvature, corneal thickness, or edema will be caused by lens wear and not by these diseases.

This work was performed in accordance with good clinical practice guidelines and the tenets of the Declaration of Helsinki [28], and it complied with the Ethics Committee (CEIC) of the San Carlos Clinical Hospital of Madrid (C.P.-C.I. 15/025-E). The participants in the study signed an informed consent form to allow the use of their clinical data for scientific purposes. It was also explained to them that they could decide to abandon the study at any time.

All patients were recruited to the study at least one year after LASIK surgery. Those who wore other contact lenses or scleral lenses prior to the study had to stop wearing them at least one month before the first assessment. All measurements were realized in both eyes, but the results were analyzed for only one eye of each subject. To avoid the direct choice of the same eye and a proprioceptive criterion, a study was performed with the random choice of each eye of the subject to avoid possible biases in the measurements performed [29,30]. On the first day, subjects underwent an eye examination, including anterior eye biomicroscope and corneal topographic analysis, employing the Pentacam HR Eye Scanner (Oculus, Wetzlar, Germany).

Each eye was fitted with an ScCL ICD, with a diameter of 16.5 mm, a center thickness of approximately 300 μm, and a 100 barrer Dk, from the manufacturer Lenticon SA (Madrid, Spain). More details about the ScCLs are in Table 1. The contact lenses were fitted by the same professional following the manufacturer’s fitting guideline. The tear reservoir thickness was measured by slit-lamp evaluation by performing an optical section to estimate the thickness of the post-lens tear layer compared to the known ScCL thickness.

Once the ScCLs were fabricated by the manufacturer and each lens was adequate for each patient, the practitioner instructed the patients on how they should use the ScCL. Unpreserved saline was used to fill the scleral lens prior to application, and the lens was immersed in peroxide every day as a daily disinfectant and cleaner.

The intention of this current study was to analyze the effect of scleral lens wear on corneal topography and corneal pachymetry over 12 months. The subjects used the ScCL for eight hours per day for one year and also attended follow-up appointments with 8 h of wear for testing. Before ScCL wear, measurements were performed at the baseline visit, and then ocular measurements were performed at the 1-month, 6-month, and 12-month follow-up visits after contact lens wear.

The anterior corneal surface curvature, posterior corneal surface curvature, corneal thickness, K flat, K steep, corneal astigmatism, and RMS over a 5 mm pupil diameter were examined before lens insertion and immediately after removing the lens at different visits with the Oculus Pentacam system (version 6.11r72). The analysis included three measurements, with an evaluation quality specification rated as good quality.

The anterior corneal curvature and posterior corneal surface curvature were examined at corneal diameters of 8, 6, 4, and 2 mm and in the meridians of 0°, 45°, 90°, 135°, 180°, 225°, 270°, and 315°. The meridian labels for left eyes were converted to the opposite, that is, 0° was nasal for all eyes, and 180° was temporary for all eyes, whether right or left. The corneal thickness was assessed in the central, temporal, nasal, inferior, and superior quadrants.

The high-contrast visual acuity (HCVA) was measured with a logarithmic visual acuity chart, ETDRS, using the Visionix VX-24 screen at 4 m. The test was performed monocularly with a photopic luminance (85 cd/m^2^). Visual acuity was measured on the first day (Day 1) with ScCL correction, immediately after ScCL insertion, and at all the follow-up appointments (1-month, 6-month, and 12-month visits).

### Statistical Analysis

SPSS 22.0 statistical software (Chicago, IL, USA) was used for the data analysis, and G*power 3 was used to calculate the estimated sample size of the study [31]. The posterior corneal curvature and anterior corneal curvature were the main variables, based on a two-sided statistically significant threshold of 0.05 and a risk of 0.20. In order to detect a difference of 0.7 units, at least 19 subjects had to be included to establish statistical significance. As a result, 20 participants were recruited to participate in the study.

To determine the normal distribution of the variables, the Shapiro–Wilk normality test was used. Changes between the topographic anterior and the posterior corneal curvature parameters in multiple comparisons between more than two assessments were evaluated by ANOVA with Bonferroni correction (Bonferroni post hoc test) for paired samples. Changes between the corneal thickness parameters prior to lens wear and after lens wear at different visits were evaluated using Student’s *t*-test for related samples, and multiple comparisons among four visits were analyzed by ANOVA for repeated measures. The mean ± standard deviation (SD) was calculated, as were the mean differences between the lens dispensing visit and the other visits. For statistically significant differences, the *p*-value was <0.05.

## 3. Results

The mean age of the participants was 43.3 ± 7.97 years, with a range of 25 to 52 years. Twelve men and eight women participated in the study. The mean flat keratometric meridian was 39.47 ± 4.43 D. The subjects wore the ScCL eight hours per day for one year. The corneal thickness changes in different corneal quadrants after ScCL wear are shown in Table 2.

The differences among the visits in all corneal quadrants were not statistically significant. After one month, the corneal thickness showed an increase without a statistically significant difference compared to the visit before wearing the ScCL. There was, however, a statistically significant increase (*p* < 0.05) in the inferior corneal thickness after 6 months compared to the baseline visit. The 12-month corneal thickness with the ScCL presented an increase in all corneal quadrants but was only statistically significant (*p* < 0.05) in the superior quadrant.

The difference values in the anterior and posterior surface corneal curvature before and after 1, 6, and 12 months of post-LASIK subjects wearing an ScCL are graphically represented in Figure 1. Globally, the anterior and posterior corneal curvature show a flattening and a steepening, respectively. In the one-month visit, a statistically significant flattening in the anterior surface (*p* < 0.05) was observed at the 0 mm, 2 mm, and 4 mm radii in all meridians and at 6 mm in the 180°, 225°, 270° and 315° meridians. The posterior surface presents a statistically significant steepening (*p* < 0.05) at the central radius and at 6 mm in the 315° meridian.

After 6 months of ScCL wear, the post-LASIK subjects showed a statistically significant anterior surface flattening (*p* < 0.05) in the central corneal radius, at the central radius; at 2 mm in all meridians; and at 4 mm in the 45°, 90°, 135°, 180°, 225°, 270°, and 315° meridians. Conversely, the posterior surface presented a statistically significant steepening (*p* < 0.05) in the central-inferior area, at the central radius; at 2 mm in the 0°, 45°, 135°, 270°, and 325° meridians; at 4 mm in the 0°, 225°, and 315° meridians; and at 6 mm in the 0°, 270°, and 325° meridians.

The anterior and posterior surface corneal curvature differences before and after 12 months of ScCL wear in the post-LASIK ectasia subjects are graphically represented in Figure 1. The values are higher than those found in the previous visits. In the anterior corneal curvature, there was a statistically significant flattening (*p* < 0.05) at the central corneal radius, at the 0, 2, and 4 mm radii. However, there was a statistically significant steepening (*p* < 0.05) at the peripheral corneal radius, at the 6 and 8 mm radii. The anterior changes present a statistically significant flattening (*p* < 0.05) at the central radius; at the 2 mm radius in the 0°, 45°, 90°, 180°, 225°, 270°, and 315° meridians; and at the 4 mm radius in the 0°, 90°, 180°, 225°, 270° and 315° meridians. Significant steepening (*p* < 0.05) appeared at 6 mm in the 0°, 45°, 90°, and 135° meridians; and at 8 mm in the 0°, 45°, 90°, 135°, 225°, 270°, and 315° meridians.

On the other hand, the posterior curvature showed a generally corneal steepening. There was significant steepening at the central area, at the central radius; at 2 mm in 45°, 90°, 135°, 180°, 225°, 270°, and 315° meridians; at 4 mm in the 45°, 90°, 135°, 270°, and 315° meridians; and at the 6 mm radius in the 315° meridian.

Table 3 describes the results obtained in the main central corneal radius, the corneal astigmatism, and the RMS. As for the K flat results, significant corneal flattening was shown after one month of use. In the K steep, there was a generalized flattening at follow-up, and it was statistically significantly flatter after the first month and at six months. As a result, corneal astigmatism was reduced during follow-up visits and was significant after one month of ScCL wear. In contrast, the RMS values decreased significantly overall and at the six-month visit and after one year.

Figure 2 shows the log-MAR high-contrast visual acuity (HCVA) at the baseline visit and after ScCL wear at all appointments. Visual acuity improved with ScCL after some months of wear, but there were no statistically significant differences among the visits (*p* > 0.05).

## 4. Discussion

Progressive post-LASIK ectasia can be shown in two forms: the first with low, irregular, and well-corrected astigmatism produced by a central ectasia, and the second as a keratoconus type, with paracentral thinning, irregular astigmatism, and low visual acuity, corrected with spectacles. This second form is of major concern, as it requires refractive correction, such as an ScCL.

Some of the reasons for fitting an ScCL are applications for corneal irregularities with various etiologies and for improving dry eye symptoms, including post-refractive surgery, which covers both applications [20].

In this study, all patients were fitted with an ScCL after LASIK surgery to achieve improved visual quality and ocular comfort, thanks to its excellent centration, low movement, and post-lens tear film [32]. These ScCL advantages are well known; however, the effect of an ScCL on ocular surface changes in this type of wearer has not been reported.

Currently, it is known that ScCLs can induce adverse effects after extended wear. One of these might be increased corneal thickness, which is indicative of edema induced by hypoxic stress. The potential effects of ScCLs on hypoxia and corneal edema have been reported [33].

In addition, other works have shown the effects of ScCLs on the corneal surface in healthy patients with regular corneas, evaluating the corneal thickness [34], anterior corneal curvature [35], and posterior curvature [36]. Nevertheless, it cannot be assumed that their results are similar for the cases of irregular cornea patients, and there are other studies that have analyzed the changes in the corneal thickness [25], anterior curvature [26], and posterior corneal curvature [27] after ScCL wear in irregular cornea subjects. However, none of these previous studies have specifically quantified the long-term changes in corneas with post-LASIK ectasia.

This study is the first to show the quantified differences in the anterior corneal surface curvature, corneal thickness, and posterior surface corneal curvature after one-year of ScCL wear in post-LASIK ectasia patients.

The present research work reports significant differences in the cornea thickness in post-LASIK subjects at different visits after ScCL wear. After six months of ScCL wear, a significant difference in the corneal thickness in the inferior area was found, with an increase of 2.13%. And after one year, the significant increase was 1.30% in the superior area. These increments in the superior and inferior regions could be due to the eyelid’s pressure over the lens, and therefore, over the cornea. Currently, no consensus has been reached on the corneal thickness change after ScCL wear in regular and irregular cornea, since some authors found an increase the corneal thickness of no more than 2% in the central cornea region [25,36], while other authors found a corneal thinning after wearing the lens [35]. However, another study showed an increase of 2.3% of corneal thickness after one year of using an ScCL in irregular cornea subjects [37], which is approached by the results of this study and statistically significant.

Considering the outcomes of the anterior surface, a flattening and a steepening in the central radius and the peripheral radius, respectively, were found. Differences between before and after ScCL wear presented higher values at the one-year visit compared to the previous visits. This difference may be related to the continued time of ScCL wear and to the biomechanical interfiber disruption that occurs in a cornea that has undergone surgical changes [13].

The values of flattening found in the anterior corneal power could cause visual clinical changes after several hours of lens wear, especially after ScCL removal. If the millimeter (mm) values are extrapolated to diopters (D) (0.20 mm equivalent to 1D), clinical flattening changes of up to approximately 3 D were found in the central cornea region, whilst the flattest change that occurred was 0.67 mm after one year of wear. However, no statistical changes in the HCVA outcomes were found during the follow-up visits of ScCL wear, but it is not known whether these changes affected vision without ScCL after their use.

Although no significant changes were found with ScCL wear for 8 h a day for one year, wearing them more hours a day and for a period longer than one year could significantly affect vision and safety and cause complications, such as increased edema, and thus, hypoxia.

These findings recommend performing topographic and quality of vision controls more regularly in this type of patient affected by surgical changes than in other patients indicated for ScCLs. This will make it possible to understand possible visual changes and patient complaints.

With respect to these results, in other scientific studies, a flattening of the anterior surface of the cornea was found after the short-term use of ScCLs in healthy individuals [35] and keratoconic eyes [26] but with smaller difference values. Nevertheless, these short-term studies cannot be extrapolated to long-term results with other subjects.

The posterior surface showed a generalized steepening of the entire cornea radius and significant differences in the central area. A possible reason could be the relationship between the disorder’s biomechanical central cornea and the place where the laser was used, since other studies described a minimal change had been observed immediately following lens removal in healthy subjects [35,36] and irregular cornea [27]. Unlike changes in the anterior corneal curvature of the cornea, differences in the posterior corneal curvature are not considered clinically relevant in terms of visual impairment [27]. This is because the corneal refractive index (1.376) and the aqueous humor refractive index (1.336) are similar, so from an optical point of view, the posterior corneal curvature differences do not affect vision. However, changes in the anterior curvature of the cornea may be clinically relevant because they may cause a change in corneal astigmatism and affect visual acuity [26].

To enhance the clinical significance of this study, corneal astigmatism and higher-order aberrations RMS were examined before and after ScCL wear. After lens removal, a decrease in corneal astigmatism was demonstrated, in concordance with a flattening of the central corneal radius and an improvement in the RMS aberration values. After one year of wear, the RMS was reduced by 50%, indicating an improvement in visual quality for the duration of the lens’ wear. Kumar et al. [38] also reported a greater than 50% reduction in the RMS after ScCL wear in post-LASIK subjects, similar to these results, but in contrast to this study, the diameter size analyzed was 4.5 mm and the sample size was a smaller number of participants.

Currently, there is no scientific literature published on refractive surgery subjects wearing ScCLs and their corneal surface changes, and therefore, it is not possible to discuss it with other similar studies. Moreover, this work presents some limitations that must be considered. The post-refractive surgery subjects were operated on using the LASIK technique, but some subjects were eight years ago and others a year ago. The time that has passed after surgery could be a variable to influence the results due to biochemical conditions. Another interesting parameter would be to know the number of diopters to eliminate in the surgery that could also influence corneal biomechanics alteration. Thus, in future studies, it would be interesting to raise the number of samples, differentiate several groups, and compare these outcomes with a healthy cornea group.

Regarding the evaluation of the correlation between corneal reversibility, it would be interesting to evaluate the corneas some hours/days after lens removal to analyze if the changes were transient (and potentially motivated by scleral lens wear) or were true changes to the corneal topography. Additionally, it would be advisable to analyze the time required for the cornea to return to its initial state after ScCL removal, and ocular refraction could be a future perspective because corneal surface changes can affect visual acuity. Scientific works about the correlations among these variables might be of interest.

In conclusion, it has been shown that the ScCL fitted in the present study significantly changed the anterior and posterior corneal curvature. The anterior surface presented a flattening and a steepening in the central radius and peripheral radius after ScCL wear, respectively, after one year of wear. The posterior surface, meanwhile, showed a significant steepening in the central region. The corneal thickness results suggest an increase in different regions and significant increments in the superior and inferior areas.

Moreover, the average high-contrast visual acuity improved with ScCL wear, and this was practically the same in all follow-up visits for a year. However, further long-term follow-up studies with more subjects and a regular cornea group are needed to understand the mechanism of post-surgery corneas.

## 5. Conclusions

ScCLs cause corneal molding in users with post-LASIK ectasia, despite not touching the cornea and resting on the sclera. The described molding could modify diopters affecting visual acuity and quality of vision. The change in corneal thickness could cause complications in the future with longer ScCL wear time, as the samples used in this study had weakened corneas with post-LASIK ectasia. The prolonged use of scleral lenses in post-LASIK subjects has been shown to cause significant changes in the corneal curvature and thickness, thus demonstrating that more detailed and periodic topographic and vision quality checks to monitor their wear in ScCL patients is recommended.

## Figures and Tables

**Figure 1 healthcare-11-02922-f001:**
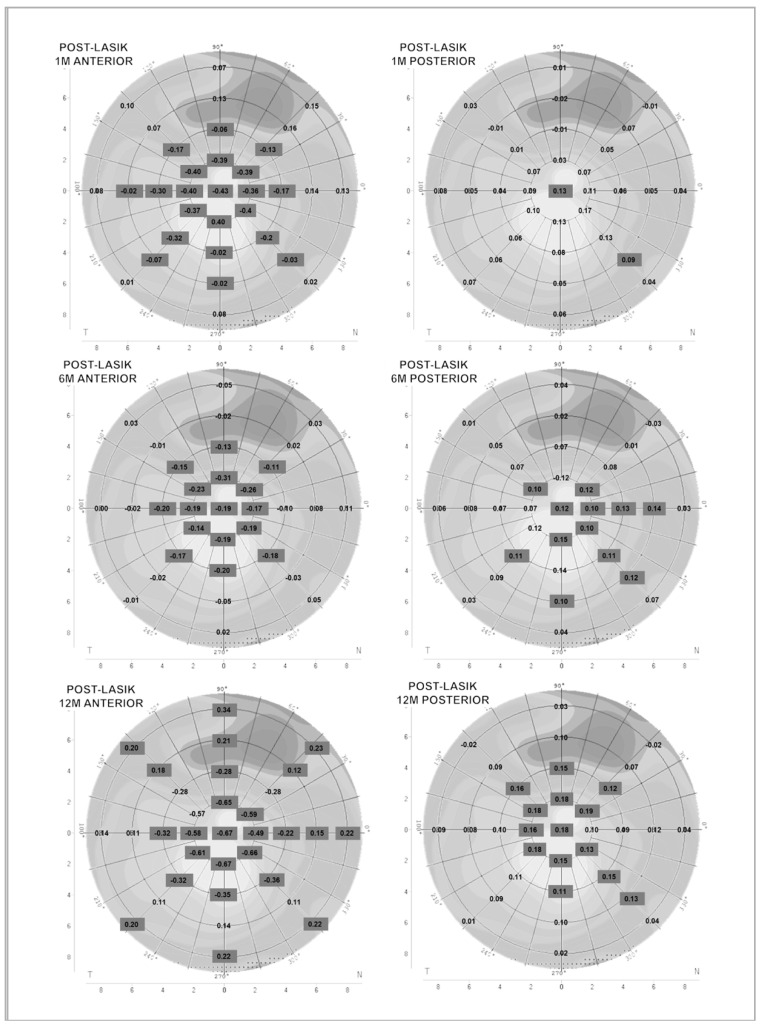
Anterior and posterior corneal curvature changes calculated in millimeters between baseline and after 1, 6, and 12 months of scleral lens (ScCL) wear in post-LASIK subjects at different positions on the cornea. The results express the mean values. Negative values mean flattening and positive values mean steepening curvature after ScCL use. Values in black boxes are statistically significant. *p* value < 0.05. ANOVA with Bonferroni post hoc corrections test.

**Figure 2 healthcare-11-02922-f002:**
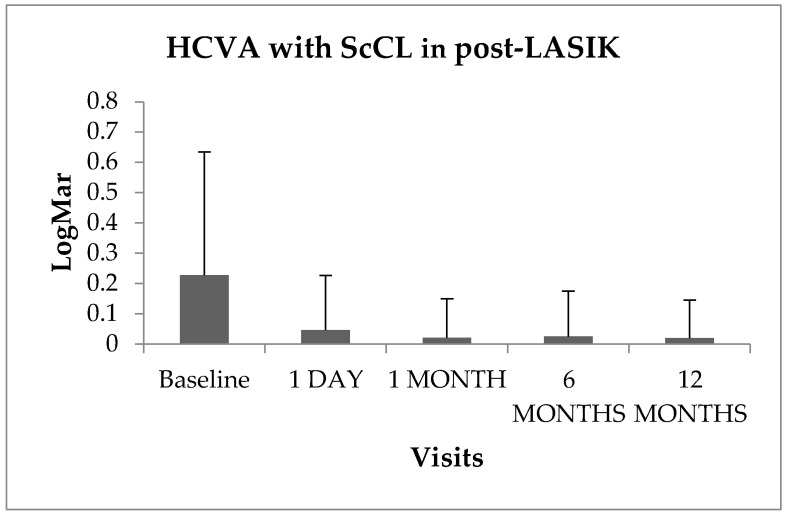
Log-MAR high-contrast visual acuity with scleral lens (ScCL) at all visits in post-LASIK subjects (0.00 LogMar = 20/20 Snellen). Error bars show mean ± SD. ANOVA test for related samples (n = 20).

**Table 1 healthcare-11-02922-t001:** Parameter details of the sclera lenses used to fit the patients in the study.

Parameters	ICD 16.5
Manufacturer	Lenticon SA
Design’s owner	KATT Inc.
Power (D)	+1.00 D to −16.00 D
Diameter (mm)	16.50
Sagittal height (microns)	3900–5600
Material (USAN)	Paflufocon D
Dk (barrer)	100
Center thickness (µm)	300
Water Content (%)	<1%

**Table 2 healthcare-11-02922-t002:** Corneal pachymetry outcomes at baseline visit without ScCL and after 1, 6, and 12 months ScCL wear in post-LASIK patients. The results are expressed as mean ± SD (n = 20). Bold numbers signify significance. * *p*-value < 0.05, Student’s *t*-test for related samples. *p*-value (1 M-6 M-12 M) < 0.05, ANOVA for repeated measures.

Corneal Thickness	Visit	1 Month Post-Lasik	6 Months Post-Lasik	12 Months Post-Lasik	*p*-Value (1 M-6 M-12 M)
Center thickness (µm) Mean ± SD	Baseline	456.91 ± 85.70			0.915
	After-ScCL	459.70 ± 89.53	466.11 ± 97.51	457.63 ± 93.46	
	*p*-value	0.970	0.674	0.801	
Nasal thickness (µm) Mean ± SD	Baseline	623.45 ± 65.50			0.982
	After-ScCL	620.10 ± 81.69	624.88 ± 69.80	623.57 ± 76.72	
	*p*-value	0.981	0.288	0.865	
Temporal thickness (µm) Mean ± SD	Baseline	602.54 ± 59.95			0.916
	After-ScCL	605.90 ± 69.41	601.66 ± 69.69	607.72 ± 62.15	
	*p*-value	0.800	0.339	0.171	
Inferior thickness (µm) Mean ± SD	Baseline	614.63 ± 39.49			0.843
	After-ScCL	613.70 ± 44.57	627.77 ± 43.55	619.27 ± 44.05	
	*p*-value	0.982	**0.024 ***	0.357	
Superior thickness (µm) Mean ± SD	Baseline	671.81 ± 34.52			0.786
	After-ScCL	676.55 ± 52.51	674.66 ± 41.80	680.54 ± 40.35	
	*p*-value	0.689	0.366	**0.043 ***	

**Table 3 healthcare-11-02922-t003:** K flat, K steep, corneal astigmatism, and higher-order aberrations (RMS) outcomes at baseline visit without ScCL and after 1, 6, and 12 months of ScCL wear in post-LASIK patients. The results are expressed as the mean ± SD (n = 20). Bold numbers signify significance. ** *p*-value < 0.05, Student’s *t*-test for related samples. * *p*-value (1 M-6 M-12 M) < 0.05, ANOVA for repeated measures.

Parameters	Visit	1 Month Post-Lasik	6 Months Post-Lasik	12 Months Post-Lasik	*p*-Value (1 M-6 M-12 M)
K Flat (mm) Mean ± SD	Baseline	9.06 ± 1.10			0.079
	After ScCL	9.14 ± 1.15	9.10 ± 1.29	9.09 ± 1.14	
	*p*-value	**0.014 ***	0.230	0.190	
K Steep (mm) Mean ± SD	Baseline	8.57 ± 1.08			**<0.001 ****
	After ScCL	8.70 ± 1.17	8.63 ± 1.11	8.60 ± 1.12	
	*p*-value	**0.006 ***	**<0.009 ***	0.204	
Corneal Astigmatism (D) Mean ± SD	Baseline	2.18 ± 1.16			0.298
	After ScCL	1.97 ± 1.23	2.13 ± 1.24	2.09 ± 1.07	
	*p*-value	**0.033 ***	0.726	0.422	
RMS (µm) Mean ± SD	Baseline		1.94 ± 0.95		**0.002 ****
	After ScCL	1.88 ± 0.92	1.15 ± 0.99	0.98 ± 0.48	
	*p*-value	0.788	**0.032 ***	**0.001 ***	

## Data Availability

Data are available from the authors upon reasonable request.

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
