# Peer review of "Anterior, Posterior, and Thickness Cornea Differences after Scleral Lens Wear in Post-LASIK Subjects for One Year"

_healthcare, 2023, doi:10.3390/healthcare11222922_

Round 1
Reviewer 1 Report
Comments and Suggestions for Authors
1. The clinical significance of this study is not clear.
2. In lines 23 and 320, please specify what "reviews" are recommended. In addition, it would be better to emphasise the clinical significance.
3. In line 37, "[1,2,5]" indicates which part of the quote?
4. Based on the Introduction and Discussion, it seems that the study is primarily focused on ectasia; line 79 states that subjects with irregular cornea are the subject. This discrepancy is puzzling.
5. Regarding line 85, please state in the text why the diseases you have specified are included in the exclusion criteria.
6. Regarding lines 93-95, do you mean that the subject had a mixture of right and left eyes? To reduce bias, it would be better if the subject was only the right eye (or only the left eye).
7. Regarding lines 137-139, it would be better to indicate the calculation and results of the estimated sample size of this study.
8. The text in lines 286-289 is confusing: please make the difference between the impact of the anterior corneal curvature and the posterior corneal curvature clearer.
Author Response
We wish to express our gratitude for the effort to review this article. We believe we have responded and modified the text in accordance with your valuable comments. We proceed to respond to or comment on all points raised. Comments and changes to the manuscript are indicated in red for the reviewer.
- The clinical significance of this study is not clear.
Thank you for your comment. The clinical significance of this study is that ScCL causes corneal molding in wearers with post-LASIK ectasia, despite not touching the cornea and resting on the sclera. The described molding could modify the diopters affecting visual acuity and vision quality. The change in corneal thickness could cause complications in the future with longer wearing time of the ScLC, as the sample used has a weakened cornea with post-LASIK ectasia.
- In lines 23 and 320, please specify what "reviews" are recommended. In addition, it would be better to emphasise the clinical significance.
Thank you for your comment. The lines 23 and 320 have been modified as follows: “more detailed and periodic topographic and vision quality checks to monitor the wear for ScCL patients”.
- In line 37, "[1,2,5]" indicates which part of the quote?
Thank you for your comment. Each quotation has been placed in its respective place: “Some of the treatments used are: rigid gas-permeable contact lenses [1,5], corneal crosslinking (CXL) [2,6], crosslinking with simultaneous topography-guided photorefractive keratectomy [7], implantations of intrastromal corneal ring segment (ICRS) [8,9], and corneal transplantation [10].”
- Based on the Introduction and Discussion, it seems that the study is primarily focused on ectasia; line 79 states that subjects with irregular cornea are the subject. This discrepancy is puzzling.
Thank you for your comment. Corneal ectasia is classified within the types of irregular corneas. To clarify the existing discrepancy, the type of irregular cornea has been specified by modifying the sentence as follows: “The patients presented corneal surface irregularities, specifically cornea ectasia after they had undergone LASIK surgery for the correction of refraction”.
- Regarding line 85, please state in the text why the diseases you have specified are included in the exclusion criteria.
Thank you for your observation. The exclusion criteria used were atopy, allergies and ocular diseases, because they could cause modifications in the ocular parameters analyzed. By avoiding these exclusion criteria, changes in corneal curvature, corneal thickness or edema will be caused by lens wear and not by these diseases.
- Regarding lines 93-95, do you mean that the subject had a mixture of right and left eyes? To reduce bias, it would be better if the subject was only the right eye (or only the left eye).
Thank you for your comment. To avoid the direct choice of the same eye and a proprioceptive criterion, a study was performed with the random choice of each eye of the subject to avoid possible biases in the measurements performed.
- Regarding lines 137-139, it would be better to indicate the calculation and results of the estimated sample size of this study.
Thank you for your comment. The corresponding information has been added:
“G*power 3 was used to calculate the estimated sample size of the study [29]. Posterior corneal curvature and anterior corneal curvature were the main variables, based on a two-sided statistically significant threshold of 0.05 and a risk of 0.20, in order to detect a difference of 0.7 units, at least 19 subjects were needed to be included to establish sta-tistical significance. As a result, 20 participants were recruited to participate in the study.”
- The text in lines 286-289 is confusing: please make the difference between the impact of the anterior corneal curvature and the posterior corneal curvature clearer.
Thank you for your comment. The difference has been clarified with the following explanation in the text: “Unlike changes in the anterior corneal curvature of the cornea, differences in the pos-terior corneal curvature are not considered clinically relevant in terms of visual im-pairment. This is because the corneal refractive index (1.376) and the aqueous humor refractive index (1.336) are similar, so from an optical point of view, the posterior cor-neal curvature differences do not affect vision. However, changes in the anterior curvature of the cornea may be clinically relevant because they may cause a change in corneal astigmatism and affect visual acuity.

Reviewer 2 Report
Comments and Suggestions for Authors
Careful review conducted on the study focusing on the analysis of anterior and posterior corneal surface shape and corneal thickness differences before and after scleral lens (ScCL) wear in post-LASIK ectasia subjects over the course of one year. Study investigates essential parameters, including corneal thickness at various locations, anterior and posterior corneal surfaces, and high-contrast visual acuity, which are pertinent to the field. This paper is very well written and easy to read. The background and rationale for the study is well-articulated. Methodology is sound.
Findings are noteworthy, particularly the statistically significant increments in corneal thickness observed in the inferior and superior areas after 6 and 12 months of wearing ScCL. Additionally, observed changes in anterior and posterior corneal curvature are significant, as they contribute valuable insights into the effects of prolonged ScCL use on post-LASIK subjects. Importantly, note that despite these changes, high-contrast visual acuity with ScCL correction remained consistent, highlighting the clinical implications of the research.
Conclusions regarding the need for more detailed and regular reviews of ScCL wear in post-LASIK patients are well-founded and carry significant implications for clinical practice. Study adds to the body of knowledge in this area and underscores the importance of monitoring corneal changes in patients utilizing ScCL.
Overall, research is a valuable contribution to the field, and the recommendation is for its publication in the scientific journal. Appreciate the thoroughness and relevance of the study, and it is believed to be of interest to the readership.
Author Response
Thank you very much for your kind comments. Your opinion has been especially gratifying for the authors. Thank you again.
Reviewer 3 Report
Comments and Suggestions for Authors
(1) To enhance the clinical significance of this study, it is recommended that the authors provide data on the changes in corneal astigmatism and higher-order aberrations after scleral lens wear in post-LASIK subjects.
(2) As stated by the author, an increase in corneal thickness may indicate hypoxic stress-induced edema. The results in this study showed a statistically significant increment of corneal thickness (p<0.05) in the inferior area after 6 months and in the superior area in the 12 months’ follow-up wearing ScCL. So, what is the corneal edema status in subjects involved in this study?
(3) Page 3 lines 93 - 95: Please clarify the process of how random selection was conducted.
(4) Page 7 lines 197 - 198: the KC subjects ? The description here is quite confusing to me.
Comments on the Quality of English LanguageThis study could benefit a lot from further language editing and revision.
Author Response
We wish to express our gratitude for the effort to review this article. We believe we have responded and modified the text in accordance with your valuable comments. We proceed to respond to or comment on all points raised. Comments and changes to the manuscript are indicated in green for the reviewer.
1. To enhance the clinical significance of this study, it is recommended that the authors provide data on the changes in corneal astigmatism and higher-order aberrations after scleral lens wear in post-LASIK subjects.
Thank you for your recommendation. Corneal astigmatism data has been added along with Ks (K flat and K steep) and higher-order aberrations with RMS. A table 3 has been included and the corresponding notes have been described in the manuscript.
2. As stated by the author, an increase in corneal thickness may indicate hypoxic stress-induced edema. The results in this study showed a statistically significant increment of corneal thickness (p<0.05) in the inferior area after 6 months and in the superior area in the 12 months’ follow-up wearing ScCL. So, what is the corneal edema status in subjects involved in this study?
Thank you for your comment and question. The state of corneal edema in the study subjects is not clinically relevant despite the significant increase in corneal thickness in the inferior and superior area. Biomicroscopic observation showed no signs of edema, and visual acuity analysis showed no significant changes. Nevertheless, it is advisable to perform a control in these ScCL users.
3. Page 3 lines 93 - 95: Please clarify the process of how random selection was conducted.
Thank you for your comment. To avoid the direct choice of the same eye and a proprioceptive criterion, a study was performed with the random choice of each eye of the subject to avoid possible biases in the measurements performed. This has been clarified in the text.
4. Page 7 lines 197 - 198: the KC subjects ? The description here is quite confusing to me.
Thank you for your appreciation. It was a mistake. It has been modified in the text by: Post-LASIK ectasia subjects.

Round 2
Reviewer 1 Report
Comments and Suggestions for Authors
Response 1. Your response would be better reflected in Conclusions section.
Response 2-5. OK.
Response 6. It would be better to describe it in the same detail as your response text and add a reference to the citation to that text, instead of modifying it as in line 98.
Response 7. OK.
Response 8. It would be better to add the references to the revised text.
Author Response
Response 1. Your response would be better reflected in Conclusions section.
Thank you for your recommendation. The following information has been added to the conclusion: “ScCL causes corneal molding in users with post-LASIK ectasia, despite not touching the cornea and resting on the sclera. The described molding could modify diopters affecting visual acuity and quality of vision. The change in corneal thickness could cause complications in the future with longer ScLC wear time, as the sample used in this study has a weakened cornea with post-LASIK ectasia.”
Response 6. It would be better to describe it in the same detail as your response text and add a reference to the citation to that text, instead of modifying it as in line 98.
Thank you for your recommendation. More detailed information and two references have been added. “To avoid the direct choice of the same eye and a proprioceptive criterion, a study was performed with the random choice of each eye of the subject to avoid possible biases in the measurements performed [29,30].”
Response 8. It would be better to add the references to the revised text.
Thank you for your comment. References have been added to the revised text.

Reviewer 3 Report
Comments and Suggestions for Authors
The manuscript has been improved.
Author Response
Thank you very much for your kind comment.